# Practice and System Factors Impact on Infection Prevention and Control in General Practice during COVID-19 across 33 Countries: Results of the PRICOV Cross-Sectional Survey

**DOI:** 10.3390/ijerph19137830

**Published:** 2022-06-26

**Authors:** Claire Collins, Esther Van Poel, Milena Šantrić Milićević, Katica Tripkovic, Limor Adler, Torunn Bjerve Eide, Liubove Murauskiene, Adam Windak, Katarzyna Nessler, Bernard Tahirbegolli, Sara Willems

**Affiliations:** 1Research Centre, Irish College of General Practitioners, D02 XR68 Dublin, Ireland; 2Department of Public Health and Primary Care, Ghent University, 9000 Ghent, Belgium; esther.vanpoel@ugent.be (E.V.P.); sara.willems@ugent.be (S.W.); 3Faculty of Medicine, University of Belgrade, 11000 Belgrade, Serbia; milena.santric-milicevic@med.bg.ac.rs; 4Institute of Public Health Belgrade, 11000 Belgrade, Serbia; katica.tripkovic@zdravlje.org.rs; 5Department of Family Medicine, Sackler Faculty of Medicine, Tel Aviv University, Tel Aviv 6997801, Israel; limchuk@gmail.com; 6Department of General Practice, University of Oslo, 0318 Oslo, Norway; torunnbjerveeide@gmail.com; 7Public Health Department, Faculty of Medicine, Vilnius University, LT-01513 Vilnius, Lithuania; murauskiene@mtvc.lt; 8Department of Family Medicine, Jagiellonian University Medical College, 31-061 Krakow, Poland; mmwindak@cyf-kr.edu.pl (A.W.); katarzynanessler@gmail.com (K.N.); 9Management of Health Institutions and Services, Heimerer College, 10000 Prishtina, Kosovo; bernardtahirbegolli@gmail.com

**Keywords:** infection prevention and control, COVID-19, general practice/family medicine, health system, organizational, interventions

## Abstract

Infection prevention and control (IPC) is an evidence-based approach used to reduce the risk of infection transmission within the healthcare environment. Effective IPC practices ensure safe and quality healthcare. The COVID-19 pandemic highlighted the need for enhanced IPC measures and the World Health Organization (WHO) emphasized the need for strict adherence to the basic principles of IPC. This paper aims to describe the IPC strategies implemented in general practice during the COVID-19 pandemic and to identify the factors that impact their adoption. Data were collected by means of an online self-reported questionnaire among general practices. Data from 4466 practices in 33 countries were included in the analysis. Our results showed a notable improvement in IPC during COVID-19 with more practices reporting that staff members never wore nail polish (increased from 34% to 46.2%); more practices reporting that staff never wear a ring/bracelet (increased from 16.1% to 32.3%); and more practices using a cleaning protocol (increased from 54.9% to 72.7%). Practice population size and the practice payment system were key factors related to adoption of a) range of IPC measures including patient flow arrangements and infrastructural elements. An understanding of the interplay between policy, culture, systemic supports, and behavior are necessary to obtain sustained improvement in IPC measures.

## 1. Introduction

In the early stages of the COVID-19 pandemic, attention and investment focused on increasing hospital surge capacity. However, in the phase of increased transmission, the demand for health services at primary healthcare facilities has escalated. As a result, primary care has been a key element of the health system response to the COVID-19 pandemic, performing tasks such as the early identification of COVID-19 patients, managing mild and moderate cases, and conducting vaccination [1]. Therefore, the effective implementation of infection prevention and control (IPC) measures in primary healthcare facilities to prevent the transmission of the virus which causes COVID-19 is a priority.

IPC is an evidence-based approach used to reduce the risk of infection transmission within the healthcare environment [2]. Effective IPC practices ensure safe and quality healthcare, as they protect both patients and health workers from healthcare-associated infection [3]. The basic principles of IPC (i.e., standard precautions including practices such as hand hygiene, use of personal protective equipment, respiratory hygiene/cough etiquette, environmental cleaning, safe injection practices, and decontamination of medical devices) are universally relevant to all healthcare settings and should be applied at all times in the care of all patients [4,5,6]. In addition, in times of crisis, such as an infectious disease outbreak, health facilities must take appropriate additional preventive measures alongside standard precautions, i.e., transmission-based precautions [5,6].

The COVID-19 pandemic has had a major impact on primary care including reduced capacity and access of patients to primary care, reduced quality of care, delays in medical treatment of non-COVID patients, rapid transition to telemedicine, and the need to provide adequate measures of IPC [7]. The pandemic highlighted the need for IPC measures, including washing hands, cleaning surfaces, ventilation of rooms, and precautions to break chains of transmission [8]. Other aspects of IPC include personal protective equipment, rapid diagnosis, physical distancing, isolation, investigation, and follow-up of close contacts [9]. Primary care physicians in Singapore identified the availability of personal protective equipment and infection prevention guidelines as priority requirements [10]. Healthcare-associated infections (HCAIs) are a significant problem for patient safety and prevention must be a priority in healthcare settings [11]. Approximately five million HCAIs are estimated to occur in acute care hospitals in Europe annually, with a corresponding economic burden of EUR 13–24 billion [11]. Studies demonstrated increases in healthcare-associated infections in 2020, indicating the need to strengthen IPC practices during the COVID-19 pandemic [12]. The World Health Organization emphasized the need for strict adherence to the basic principles of IPC in all healthcare facilities and outlined the minimum requirements for IPC in primary healthcare institutions adopted for the context of COVID-19 [6,13,14].

In line with the World Health Organization’s (WHO) recently published document on strengthening IPC in primary care, hand hygiene improvement prioritizes the sense of resource allocation within the IPC program [15]. To prevent the healthcare-associated infection caused by COVID-19, it is essential that both staff and patients adhere to proper hand hygiene as a leading infection control measure used in response to viral outbreaks [16]. Although a supply of clean running water in most European countries is a standard, the other weak points such as a lack of alcohol-based hand rubs or hand sanitizers at each point of care may be present and jeopardize IPC [17]. WHO also recommends not wearing jewelry rings or bracelets as well as keeping nails unvarnished and short in healthcare settings [18].

This paper aims to describe the infection control measures implemented in general practice during the COVID-19 pandemic and to identify the factors that impact their adoption. Our theoretical basis is that while IPC improvements across countries will have been achieved during the pandemic, systemic factors impact their adoption.

## 2. Materials and Methods

### 2.1. Study Design and Setting

In the summer of 2020, an international consortium of more than 45 research institutes was formed under the coordination of Ghent University (Belgium) to set up the study to consider how primary care practices were organized during the COVID-19 pandemic (PRICOV-19). This multi-country cross-sectional study specifically focused on quality and safety in primary care practices during the COVID-19 pandemic. Data were collected in 37 European countries and Israel. Data were collected by means of an online self-reported questionnaire among general practices with one questionnaire returned per practice. The questionnaire was developed at Ghent University in multiple phases, including a pilot study among 159 practices in Flanders (Belgium). More details are described in the protocol [19]. The questionnaire consisted of 53 items divided into six topics: (a) infection prevention; (b) patient flow for COVID and non-COVID care; (c) dealing with new knowledge and protocols; (d) communication with patients; (e) collaboration; (f) wellbeing of the respondent; and (g) characteristics of the respondent and the practice. The questionnaire was translated into 38 languages following a standard procedure. The Research Electronic Data Capture (REDCap) platform was used to host the questionnaire in all languages, send out invitations to the national samples of practices, and securely store participants’ answers [20]. 

### 2.2. Sampling and Recruitment

The data reported here were collected between November 2020 and December 2021, except for Belgium, where data were partially collected earlier. Data collection varied between countries from three to 35 weeks. In each partner country, the consortium partner(s) recruited general/family practices following a pre-defined recruitment procedure [19]. There was no funding for this study and coordinators recruited practices out of good will; however, a randomized sample was requested where possible. One questionnaire was completed per practice, preferably by a general practitioner/family practitioner or by a staff member familiar with the practice organization. In most countries, the number of practices was unknown, although the number of individual practitioners may have been available. Given that the voluntary nature of participation and the number of practices were unknown, it was not possible to enforce either a specific recruitment strategy or specific response numbers/rate. The majority of countries chose a sample from the entire population of general practices with this being a convenience sample for approximately one half (Table 1). At least one reminder was sent in all countries. The response rate varied and generally targeted convenience samples which attracted larger response rates, likely due to targeting personal contacts or practices within the lead investigator’s own area. However, some notably high response rates from some random/convenience national samples (e.g., Bulgaria, Greece, Serbia) were also noted (Table 1).

### 2.3. Data Management and Analysis

Ghent University was responsible for the data cleaning and running consistency checks on the international data using a detailed data management plan and protocol [19]. 

As all questions were options, the number of responses to each individual question varied substantially. A seven-item infection control infrastructure equipment (ICIE) score was created based on having each of the seven infection control items in every consulting room (sink, non-contact tap, non-contact bin, disposable gloves, disposable coats, surface disinfectant, and paper cover for examination table). For this paper, cases missing data on all seven infection control equipment variables were excluded from the analysis in this paper resulting in 4466 practices being included.

Statistical analysis was performed using SPSS software (version 28.0 SPSS Inc., Chicago, IL, USA) on Version 7 of the database which was the version consisting of the cleaned data of 33 countries available as of 3 November 2021. Chi-squared analysis was used to investigate the relationship between variables with *p* < 0.05 considered statistically significant.

Logistic regression was used to determine associations with infection control measures. For IPC measures during COVID-19, namely calling patients where risk status was unknown, leaving time between consultations for disinfection, and leaving home visits to the end of the shift, regularly/always were compared to sometimes/rarely/never. Regarding the reported practice limitations, those who reported a large extent of structural limitations were compared to other responses (none, hardly, a limited extent). 

The explanatory practice characteristics considered were the size of the practice population, the practice payment system, the practice location, the number of general practitioners (GPs)/GP trainees, and the number of different disciplines working in the practice. The disciplines were listed on the questionnaire, e.g., dietician, physiotherapist, etc., and they ticked to indicate if they worked in the practice or not; the total number of disciplines ticked is represented by this variable. The practice payment system used was a centrally created variable amalgamating the relevant response options in each country into three overall categories appropriate for international comparison [19]. Multi-collinearity was tested and, due to the high correlation between practice variables, the practice population size, the practice payment system, and the practice location were retained in the logistic regression models. The practice population size showed a skewed distribution and, at the data cleaning stage, outliers were confirmed with each country coordinator as valid [19]. For this analysis, data were recoded into groups.

### 2.4. Ethical Approval

The study was conducted according to the guidelines of the Declaration of Helsinki. The Research Ethics Committee of Ghent University Hospital approved the protocol of the PRICOV-19 study (BC-07617). Research Ethics Committees in the different partner countries gave additional approval if needed in that country. All participants gave informed consent on the first page of the online questionnaire. 

## 3. Results

The analysis included 4466 practices who had a valid ICIE score. Responses from 33 countries were received (Table 1). A description of the sample is shown in Table 2. Overall, 37.6% of respondents worked in single-handed practices and one fifth (24.8%) in practices with six or more GPs. The median patient population size was 2727 and the median number of professional disciplines working in the practices was three. In terms of location, 18.4% of practices were in rural areas, 20.1% in mixed urban/rural, and 42.8% were based in cities/suburbs. Almost equal proportions (circa two-fifths) had a fee for service or capitation and the remaining one fifth of practices had a mixed payment system. 

Practices were asked if every consultation room in the practice contained each of the seven items listed (Table 3). A hands-free tap was the item least often available in all consultation rooms. The sum of items of equipment variable was summarized, and this infrastructure (ICIE) score was found to have a median of 6 IQR (5–7). 

More than a half of practices (59.2%) always or regularly called patients who made an appointment if it was unclear whether they pose a risk of infection to verify this risk as a rule; 27.2% rarely or never did this (Figure 1). Using logistic regression, the practice payment system remained the only significant explanatory variable (*p* < 0.001). Among those with a fee-for-service system, 48.8% regularly/always called patients compared to 66.6% with a capitation system and 64.8% of those with a mixed system.

With regard to whether sufficient time wass provided between consultations for the disinfection of the consultation room, 65.3% stated that sufficient time wass provided always or regularly, in contrast to 21.5% reporting that it happened rarely or never (Figure 1). Using logistic regression, the practice population size remained the only significant explanatory variable (*p* < 0.001), with a decreasing proportion reporting sufficient time to disinfect rooms as the population size increases.

Overall, 70.8% reported that home visits were always or regularly organized so that potential COVID-19 patients were seen at the end of the GP round with 22% stating this happened rarely or never occurs (Figure 1). In the logistic regression, the practice population size (*p* < 0.001) and practice location (*p* = 0.002) were significant factors. Those in rural or mixed/rural areas and with smaller practice populations were more likely to report this occurred regularly/always.

One fifth of practices reported a large degree of limitations related to the building or the infrastructure of the practice to provide high-quality and safe care since the COVID-19 pandemic. A further 37.2% reported limitations to a limited extent, 20.6% hardly any limitations, and 21.4% no limitations. Using logistic regression, the practice payment system (*p* < 0.001) was significantly related to experiencing building/infrastructural limitations to a large extent. Such limitation were experienced by 11.1% of practices with a fee for service payment system, 23.2% of those with a capitation system and 36.3% of those with a mixed system. 

We identified a considerable improvement in practices with more practices reporting that staff members never wore nail polish (increased from 34% to 46.2%); more practices reporting that staff never wear a ring/bracelet (increased from 16.1% to 32.3%) during the COVID-19 pandemic period; and more practices using a cleaning protocol (increased from 54.9% to 72.7%) (Figure 2). The availability of sanitizer in consultation rooms, for house calls, and at the practice door for patients all increased from 75.3%, 60.8%, and 34% to 93%, 89.5%, and 89.1%, respectively, during the pandemic. Prior to the pandemic, 63.7% of practices did not provide a separate medical bag for home visits to patients with suspected infection, but this rate dropped to 36.4% during the pandemic period. 

## 4. Discussion

Our study has demonstrated that during the COVID-19 pandemic all examined IPC measures improved compared with the situation prior to the pandemic. Previous studies also indicated advances in implementing IPC measures and better IPC compliance among healthcare workers (HCWs), both during COVID-19 [21,22] and other viral outbreaks [23,24]. It is expected that outbreak risk drives positive changes in IPC behaviors of healthcare workers (HCWs) due to fear and heightened awareness of the importance of adhering to IPC guidelines [25]. However, of note is that many of these improvements were achieved from a very low pre-COVID-19 base and could be a result of improved support from management, clarity regarding requirements, HCWs seeing the value of IPC, and recognizing personal responsibility, as outlined in the literature [26,27]. 

Direct comparisons of our results to other studies were difficult due to differences in tools used in assessing IPC measures, examined elements of IPC practice, characteristics of participants, and settings. Additionally, limited studies on the IPC practice during emerging serious infections have been conducted at the primary care level. 

There is an urgent need for improved infection control and ensuring safe care in primary care facilities for at least three reasons. Firstly, as an entry point of the health system, primary care is on the frontline of outbreak control. Secondly, the previous viral outbreaks experience revealed non-optimal knowledge and behavior regarding infection control among practitioners at primary care level [28]. Our study confirmed this, as the respondents still have not achieved full compliance with IPC measures. Finally, studies indicated the possibility of a decline in compliance and difficulties in maintaining its level in the long term [29,30]. Communication, support from managers, workplace culture, training, and the physical space [26] are central to achieving this. There is an interplay between policy, culture, systemic support, and behavior [27], which must be understood in order to gain and sustain improvement in IPC measures.

Future studies are needed to identify barriers and facilitators of optimal implementation of IPC measures and maintain the level reached in the primary healthcare settings during outbreaks. In addition, a better understanding of all factors underpinning the behavior of HCWs at primary health facilities in relation to IPC practices might be useful in creating tailored strategies to combat both current and future infectious disease outbreaks.

### The Limitations and Strengths of the PRICOV-19 Study

Data collection took place over a relatively long period. It is a limitation that the questionnaire did not collect data on the waves and stages of COVID at the time of completion. This could have varied between countries but also within countries, particularly if data collection was over a longer period. However, it is not possible to accurately retrospectively establish the exact COVID burden at each time point in each country, and this restricts our ability to comment on the impact this may have had on the results. Our surveys are based on a self-selecting sample, which comes with inherent bias. This is often referred to as volunteer bias and can be mitigated by larger sample sizes, as we have here overall. Self-selection/volunteer bias may have resulted in either higher or lower reporting. Given the potential volunteer bias and the cross-sectional survey design [19], the direct assessment of causal relationships was not possible. Exact data on the population of general practices in every partner country was not available to calculate the target sample size and, additionally, given the volunteer nature of the study, no minimum sample size requirements were applied to participation. For this reason, we have not presented individual country-level data. Given that full randomization was not possible in all countries, a sampling bias may exist which may affect external validity. Some strategies were implemented to minimize the potential biases encompassed with conducting multi-center surveys. Each partner undertook the translation (and back-translation) and cultural adaptation of the questionnaire first and then after resolving terminology issues, the collaborators reached the harmonized version of the questionnaire, with consideration of local arrangements and definitions. This rigorous development of the questionnaire is a strength of the study. Other strengths of the study include the large sample size, the broad scope of respondents, and the inclusion of almost all different circumstances that European primary care operates under to make the findings more generalizable.

Our research revealed that the vast majority of European practices are well furnished to efficiently apply IPC measures. A relatively low percentage of general practice consultation rooms were equipped with taps operated by an elbow or movement detector, and this might be of note for GPs or practice managers planning infrastructural improvements. Even though improvements were observed, there are still large IPC guidelines which are not implemented, for example wearing nail polish, a ring, or bracelets. Our study implies that there is a need to provide practical implementation information alongside guidelines and to clearly specify the aspects that should be implemented in general practices. Tailored strategies for practices are required in addition to a better understanding of factors underpinning the behavior of HCWs.

## 5. Conclusions

In addition to behavioral factors, practice and system dependent factors impact IPC measures. These results can inform stakeholders with regards to tailoring intervention strategies and the systematic promotion of IPC in order to improve baseline infection prevention in primary healthcare and the rapid additional measures should we experience future infectious pandemics. An understanding of the interplay between policy, culture, systemic supports, and behavior is necessary to obtain sustained improvement in IPC measures. 

## Figures and Tables

**Figure 1 ijerph-19-07830-f001:**
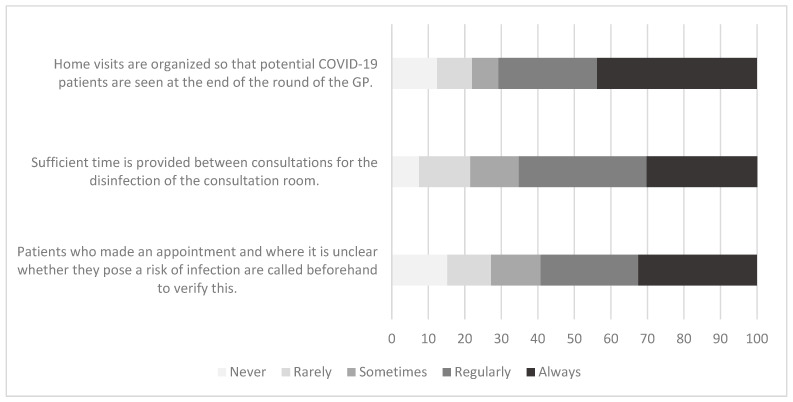
Patient flow arrangements.

**Figure 2 ijerph-19-07830-f002:**
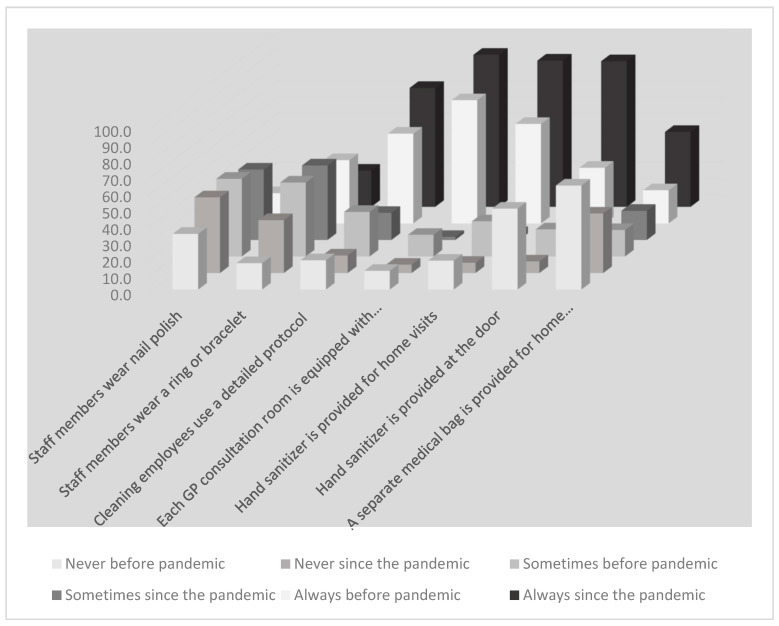
IPC measures before and since COVID-19.

**Table 1 ijerph-19-07830-t001:** Sampling method and overall response rate in each country.

	Study Invitation Sent to:	Overall Response Rate
Austria	Random National Sample	28.0%
Belgium	Random National Sample with additional convenience sample	29.7%
Bosnia and Herzegovina	Total population	5.5%
Bulgaria	Convenience National Sample	94.3%
Croatia	Convenience National Sample	11.7%
Czechia	Random National Sample from 4 regions and from list of young practitioners	22.0%
Denmark	Total population	1.5%
Estonia	Total population	13.9%
Finland	Convenience National Sample	15.5%
France	Total population	2.1%
Germany	Convenience Sample in 6 areas	15.5%
Greece	Random National Sample	94.0%
Hungary	Convenience National Sample	23.4%
Iceland	Convenience National Sample	23.8%
Ireland	Total population	12.2%
Israel	Convenience National Sample	21.8%
Italy	Convenience National Sample	25.6%
Kosovo *	Convenience Sample in 5 areas	73.3%
Latvia	Total population	9.2%
Lithuania	Convenience National Sample	22.5%
Malta	Total population	6.5%
Moldavia	Convenience Sample from 2 municipalities	24.2%
Netherlands	Random National Sample with additional convenience sample	18.9%
Norway	Total population	10.5%
Poland	Convenience National Sample	10.4%
Portugal	Random National Sample with additional convenience sample	22.9%
Romania	Convenience National Sample	25.0%
Serbia	Convenience National Sample	90.0%
Slovenia	Convenience National Sample	19.8%
Spain	Convenience National Sample	77.0%
Sweden	Convenience National Sample	7.2%
Switzerland	Convenience Sample	32.0%
Turkey	Convenience Sample	27.9%
TOTAL		27.8%

* All references to Kosovo, whether the territory, institutions or population, in this project, shall be understood in full compliance with United Nations Security Council Resolution 1244 and the ICJ Opinion on the Kosovo declaration of independence, without prejudice to the status of Kosovo.

**Table 2 ijerph-19-07830-t002:** Practice profile.

	*n*	%
**Number of GPs/GP Trainees**		
1	1484	33.2
2–3	1193	26.7
4–5	680	15.2
>5	1109	24.8
**Range of disciplines in the practice** (Median: 3; IQR: 2–5)		
1–2	1385	31.9
3–4	1493	34.3
5+	1469	33.8
**Practice Location**		
Big (inner)city	1455	32.7
Suburbs	450	10.1
(Small) town	827	18.6
Mixed urban–rural	895	20.1
Rural	819	18.4
**Practice payment system**		
Fee for service	1716	39.2
Capitation	1854	42.4
Other and/or mixed	809	18.5
**Size of practice population** (Median: 2727; IQR: 1550–7000)		
≤1500	1081	24.2
>1500–≤3000	1381	30.9
>3000–≤7000	933	20.9
>7000	1071	24.0

**Table 3 ijerph-19-07830-t003:** Availability of equipment in every surveyed GP consultation room.

Equipment Items	*n*	%
Disposable GP coats	3159	70.7
Disposable gloves	4381	98.1
Surface disinfectant	4378	98.0
Paper to cover examination table	4119	92.2
A sink	4272	95.7
A tap operated with the elbow or with a movement detector	2297	51.4
A trash can that can be operated without contact with the hand	3956	88.6

## Data Availability

The anonymized data are held at Ghent University and are available to participating partners for further analysis upon signing an appropriate usage agreement.

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
