# Peer review of "Practice and System Factors Impact on Infection Prevention and Control in General Practice during COVID-19 across 33 Countries: Results of the PRICOV Cross-Sectional Survey"

_ijerph, 2022, doi:10.3390/ijerph19137830_

Round 1
Reviewer 1 Report
The overall manuscript is neat and written concisely—with relevant information for existing literature. The quality of the work is sufficient and, therefore, I only have a few minor improvements (see below). My compliments for the great work.

Author Response
Thank you for your review - we have addressed all of your comments as outlined in the attached reply. Your contribution has improved our paper and we are grateful for same.

Reviewer 2 Report
Dear authors.
Congratulations for the effort to develop this article.
I suggest some points for improvement:
Page 1, Lines 26 to 35: Abstract: Include a short text exploring the main results found in the study;
Page 3: Create a section ‘2. Theoretical framework': To further explore the discussion of the researched theoretical basis, I suggest adding elements that contextualize the research problem;
page 3, '2. Materials and methods’: Detail the methodology construction process, explaining each step of the method until obtaining the results;
page 6, Line 226: The results are very well explained, relating to the hypotheses;
page 10, line 299: In the conclusions section, I suggest highlighting in more depth the contributions to this thematic area, as well as the opportunities arising from this study.
I hope to have contributed to the improvement of the study.
Author Response

(The authors gave the same response as above.)

Reviewer 3 Report
The manuscript titled “Practice and system factors impact on infection prevention and control in general practice during COVID-19 across 33 countries: results of the PRICOV cross-sectional survey” by Collins et al attempts at recognizing the IPC practices most implemented in healthcare settings during the Covid19 pandemic and to identify factors determining their implementation.
The study, although suffering from limitations in data collection, provides some interesting figures on which IPC practices were most or least put into practice during the pandemic. The numbers are important to highlight the areas that need to be strengthened in the healthcare setting, to control spread of infection in a similar future event.
Comments and major concerns:
The study is well conducted, given the limitations researchers faced in data collection. The data is clearly documented and well presented. However, the manuscript will benefit greatly from improving the methods section.
1) IPC measure: the authors have not provided the full name for this abbreviation in the manuscript (infection prevention and control). Please provide full name the first time this abbreviation is used in the manuscript.
2) It seems that the authors have used the term “general practitioner” and “general practice” interchangeably at multiple places in the manuscript. Please refer to a single nomenclature in the manuscript and clarify, because the term “practice” could easily be misunderstood as an IPC practice.
3) Line 126: “The response rate varies and generally targetted convenience samples attracted larger response rates, however this is not consistent with recruitment/sampling strategies as one might expect with some notable high response rate from random/convenience national samples”. Would the authors like to comment on the plausible reasons behind this inconsistency? It is important because this is an important part of the data collection strategy, and such inconsistencies could cause sampling errors.
4) Line 137: “ Ghent University was responsible for the data cleaning of the international data.” Please clarify the meaning of the term data cleaning and the methods used for the same.
5) Line 155 and 174: please clarify the meaning of the phrase “different disciplines working in the practice”. It is unclear what the authors are referring to.
Overall, the study provides some valuable insights into the IPC measures during Covid19 pandemic, which can be instrumental in marking essential areas for improvements in the existing system.
Author Response

(The authors gave the same response as above.)

Round 2
Reviewer 2 Report
Congratulations on the manuscript.
The improvements are adequate.